# Development and validation of a data quality index for forensic documentation of sexual and gender-based violence in Kenya

Rose McKeon Olson[1,2]*, Wendy Macias-Konstantopoulos[2,3], Roseline Muchai[4], Katy Johnson[4], Ranit Mishori[4,5], Brett Nelson[2,6]

1 Department of Medicine, Brigham and Women's Hospital, Boston, Massachusetts, United States of America, 2 Harvard Medical School, Boston, Massachusetts, United States of America, 3 Department of Emergency Medicine, Center for Social Justice and Health Equity, Massachusetts General Hospital, Boston, Massachusetts, United States of America, 4 Physicians for Human Rights, Boston, Massachusetts, United States of America, 5 Georgetown University School of Medicine, Washington, DC, United States of America, 6 Divisions of Global Health and Neonatology, Department of Pediatrics, Massachusetts General Hospital, Boston, Massachusetts, United States of America

* rolson@bwh.harvard.edu

**Data Availability Statement:** All relevant data are within the paper and its Supporting information files.

## Abstract

### Introduction

High-quality forensic documentation can improve justice outcomes for survivors of sexual and gender-based violence, but there are limited tools to assess documentation data quality. This study aimed to develop and validate a data quality assessment index to objectively assess clinician documentation across the 26 key elements of the standardized forensic evidence forms used in Kenya.

### Methods

Informed by prior quality assessment tools, an initial draft of the index was developed. Feedback from Kenya- and U.S.-based clinicians and human rights experts was solicited and incorporated into the draft index in an iterative fashion. Two raters independently employed the finalized Physicians for Human Rights Data Quality Index to assess and score the quality of documentation across 31 clinician-completed forms. Inter-rater reliability was determined using Cohen kappa (κ) coefficients.

### Results

The Index was found to have substantial overall reliability. Of the 26 documentation items, the Index had a *perfect* (κ = 1.0) and *almost perfect* (κ = 0.81–0.99) level of inter-rater agreement across 17 (65.4%) and 5 (19.2%) items, respectively. On a low-to-high documentation quality scale of 0 to 2, the majority of items (n = 19, 73.1%) had a mean documentation quality score >1.5–2.

### Conclusion

Quality assurance of forensic documentation is an essential component of post-sexual assault care. To our knowledge, this is the first validated quality-assessment tool in the

**Funding:** The funders had no role in study design, data collection and analysis, decision to publish, or preparation of the manuscript.

**Competing interests:** The authors have declared that no competing interests exist.

peer-reviewed literature for sexual assault documentation and may be a promising strategy to enhance the quality of sexual assault documentation in other settings, locally, regionally, and internationally.

## Introduction

Sexual and gender-based violence (SGBV) is a serious issue that affects millions worldwide, impacting people of all genders, ages, and sexual orientation. Sexual violence includes any sexual act or attempted act where consent is not obtained or freely given, often through use of violence and coercion [1]. The World Health Organization (WHO) estimates that approximately 30% of women worldwide have experienced physical and/or sexual violence by an intimate partner or non-partner in their lifetime; significant numbers of men and boys also experience sexual violence [2–3]. Rates of sexual assault are similar in Kenya [4–6], where intimate partner violence has been named one of the top ten leading risk factors driving combined death and disability [7]. Sexual violence is also a major contributor to a broad range of physical, psychological, social, legal, and economic consequences that adversely affect survivors, families, communities, and society at large [4, 5].

Survivors of sexual assault deserve timely and high-quality forensic examination, evidence collection, and documentation as part of comprehensive care for survivors. High-quality documentation of the clinical exam after sexual assault has been shown to increase trial, prosecution, and conviction rates of perpetrators [8–11]. A South African study analyzed the association of sexual assault injury documentation and legal outcomes, and found that conviction was more likely when cases had documented injuries, whether nongenital or ano-genital injuries [12]. Furthermore, an evaluation conducted in Kenya found that the relative amount of medical evidence that appeared in the Post-Rape Care (PRC) Form legal record was associated with an increased likelihood of an adjudication outcome favoring the survivor [9]. In addition to legal justice outcomes, timely evidence collection may have other positive effects, such as enhancing survivor agency, and empowering and validating the experience of survivors.

While high-quality documentation by health care professionals can improve justice for survivors, methodology to grade the quality of SGBV documentation is lacking. There are no published validated tools on quality assessment of sexual assault documentation. One non-peer reviewed index developed in South Africa found wide variability in data quality of post-rape documentation forms, depending on profession and location of data collection [13]. While problematic, this finding is a natural consequence of the wide variability in the quality, components, and professional training level of post-sexual assault evidence collection. In Kenya and in many other under-resourced contexts, there have been reports of low-quality medico-legal documentation after sexual assault [14, 15]. For example, in several contexts the sexual assault exam is heavily based on a hymen examination, which is not an accurate or reliable indicator of sexual assault [16]. These findings suggest that data quality assessments are underutilized, and their appropriate use may strengthen medico-legal evidence and thereby increase trial, prosecution, and conviction rates of perpetrators of sexual violence [10.]

The international nonprofit organization Physicians for Human Rights (PHR) partners with medical, legal, and law enforcement professionals in Kenya, the Democratic Republic of Congo, and beyond to improve the medico-legal response to sexual violence and bolster

accountability for associated crimes. PHR focuses on improving the quality and availability of forensic evidence through research, tools, and innovations. Working in close collaboration with multisectoral partners, PHR developed MediCapt, a mobile application that enables clinicians to securely document evidence of sexual violence and safely transmit and store the protected information. As part of an evaluation of the MediCapt project in Kenya, PHR worked with external evaluators to explore the option of a data quality index to eventually compare the quality of standard government-issued paper-based PRC forms with PHR's mobile MediCapt application.

The aim of this study was to develop and evaluate the PHR Data Quality Index to objectively assess the data quality and inter-rate reliability of forensic evidence documentation of sexual assault by clinicians in Kenya.

## Materials and methods

The Kenyan government's PRC Form is a two-page, triplicate form used by clinicians in Kenya to document survivor-reported sexual assault events and includes the physical examination, psychological assessment, and clinical management by the clinician. The form is divided into two sections: Part A, the description of the incident, the physical examination findings, and the documentation of the clinical management and forensic evidence; and Part B, the psychological assessment.

Informed by prior unpublished quality assessment tools [13, 17], an initial draft index was developed to objectively assess key elements of the PRC Form. We defined data quality within its six well-established dimensions: accuracy, completeness, consistency, timeliness, validity, and uniqueness [18]. The index was designed to specifically target the key components of the Kenyan government PRC Form and included a weighted scoring for each item assessed based on the quality of the data documented in the two-page paper form.

This draft index was subsequently shared in an interactive and iterative process with experienced Kenya- and U.S.-based human rights experts and clinicians. Their feedback informed a revision of the index, which was then shared with an eight-member group of Kenyan clinical, legal, and law enforcement professionals, as well as members of the PHR network in Kenya with long involvement in the care of survivors of SGBV. During a semi-structured videoconference, these professionals provided additional feedback on elements of the PRC Form most critical for documentation and prosecution of SGBV. The index, including item scoring weights, were revised accordingly.

Using the finalized PHR Data Quality Index, two reviewers (RO, BDN) independently scored each of 31 completed post-rape forms for the Index's 26 quality-metric items. All items were scored on a scale of 0 (no data or low quality), 1 (moderate quality), or 2 (high quality), with the exception of item *Part B*, the *psychological assessment*, where a scale of 0, 2, and 4 was used to place greater weight on this large component of the PRC Form and to allow for a more granular quality assessment of the psychological narrative. During initial scoring attempts, lack of clarity on some Index items was discussed by the researchers and addressed by adding a scoring guide to each of the Index items. Independent scoring was then repeated using this finalized Index.

Level of inter-rater reliability was determined with SPSS 26.0 (IBM, Armonk, NY, USA) using Cohen kappa coefficients. These coefficients were interpreted according to the following definitions: poor agreement (0.00), slight agreement (0.01–0.20), fair agreement (0.21–0.40), moderate agreement (0.41–0.60), substantial agreement (0.61–0.80), almost perfect agreement (0.81–0.99), and perfect agreement (1.00) [19].

This study was approved by the Georgetown University institutional review board in the United States. (Protocols 2016–0661 and 2016–1404) and the Egerton University institutional review board in Kenya (Approval Number EUREC/APP/099/2020).

## Results

### Inter-rater reliability of the PHR Data Quality Index

The finalized Data Quality Index (Table 1) includes 26 data quality items and is presented below.

The overall kappa score for the PHR Data Quality Index was 0.77, corresponding to a *substantial* level of agreement (Table 2). In six of the seven multi-item Index categories, independent scoring for at least half of the category items had kappa scores of 1.00, indicating a high inter-rater reliability. All items within four Index categories (*demographics*, *management*, *laboratory samples*, and *psychological assessment*) had *perfect* levels of agreement across the two independent raters, with itemized kappa scores of 1.00.

Items with a perfect agreement score (kappa 1.00) included information such as the dates of the incident, exam, and form completion; survivor name, date of birth, and contact information; perpetrator information, including body marks; information regarding care management, referrals, and laboratory studies; police officer date and signatures; and legibility. Most common errors in completing the form were reflected in items with lower agreement scores, including chief complaint, circumstances surrounding the incident, summary body map statement, and examining officer date and signature. Table 3 provides a summary of the Index's item-by-item inter-rater reliability by level of agreement.

### Applying the PHR Data Quality Index to assess quality of forensic documentation

To understand which of the Index's 26 items may be more challenging for quality documentation, a mean data quality score was determined for each Index item. With a maximum data quality score of 2, the large majority (n = 19, 73.1%) of the 26 Index items received a mean data quality score of $\geq 1.5$–2, indicating high-quality documentation by clinicians (Table 4). For the purposes of comparison, Item #24, which is typically scored out of 4, had an adjusted score of 1.81 when adjusted to a two-point scale. Four (15.4%) Index items received a mean rater score of >1.0–1.5, indicating moderate-quality documentation. These items included data on orphans and vulnerable children (OVC) status (mean = 1.48, κ = 1.00), chief complaints (mean = 1.48, κ = 0.60), circumstances surrounding the incident (mean = 1.32, κ = 0.63), and summary statements of genital exams (mean 1.44, κ = 0.96). Three (11.5%) items with low-quality mean scores $\leq 1.0$ for documentation included date of last consensual intercourse (mean = 0.84, κ = 1.00), police officer signature and date (mean = 0.13, κ = 1.00), and document signed by the examining officer within 48 hours of patient visit (mean = 0.13, κ = 1.00).

## Discussion

High-quality forensic documentation can facilitate increased investigation, prosecution, and conviction rates for survivors of sexual violence [8–10], yet no validated, published tools are available to assess the quality of documentation after sexual assault. To our knowledge, this is the first peer-reviewed study to develop a validated quality-assessment tool for sexual assault documentation. The PHR Data Quality Index had substantial inter-rater agreement, suggesting it is a valid tool to grade quality of sexual assault documentation and guide targeted

**Table 1. PHR data quality index for assessing quality of sexual violence documentation.**

| | Circle appropriate score | | |
|---|---|---|---|
| **Demographics:** | | | |
| 1. All 4 dates (dates of form, birth, exam, incident) | 0<br>No dates | 1<br>Some dates | 2<br>All dates |
| | *One point if 1–3 dates. Two points if ALL 4 dates (dates of form, birth, exam, and incident) (or "n/a").* | | |
| 2. Three names of survivor | 0<br>No names | 1<br>Some names | 2<br>All names |
| | *One point if partial name. Two points if full name.* | | |
| 3. Survivor contact info (address and phone) | 0<br>No contact info | 1<br>Only address or phone | 2<br>Both address and phone |
| | *One point if either address or phone. Two points if both address and phone (or "n/a").* | | |
| 4. OVC status | | 0<br>Not present | 2<br>Present |
| | *Two points if any status is marked.* | | |
| **History:** | | | |
| 5. Perpetrator info (gender, est. age or adult/non-adult, unknown/known) | 0<br>No info | 1<br>Some info | 2<br>All info |
| | *One point if info on 1–2 of these items. Two points if info on ALL 3 items (any info on gender, age/adult/non-adult, and perpetrator unknown/known).* | | |
| 6. Chief complaints | 0<br>No info | 1<br>Some info | 2<br>Detailed info |
| | *One point if any info, but no specific reason for visit (e.g., "patient is withdrawn"). Two points if specific reason for patient's visit (e.g., ". . .sexual assault," ". . .psychological concerns," etc.).* | | |
| 7. Circumstances surrounding incident | 0<br>No info | 1<br>Some info | 2<br>Detailed info (must include penetration & struggle info) |
| | *One point if any info. Two points if info on BOTH "penetration" and "struggle" (or "n/a").* | | |
| 8. Previous reporting and care | 0<br>No info | 1<br>Some info | 2<br>Detailed info |
| | *One point if any info. Two points if info on BOTH "previous reporting" and "previous care" (or "n/a").* | | |
| **Physical examination:** | | | |
| 9. Notations on body map | | 0<br>Not present | 2<br>Present (or marked "normal" or similar) |
| | *Two points if any notations or marked "normal" or "n/a."* | | |
| 10. Statement in "Comments" summarizing body map exam | 0<br>No info | 1<br>Some info | 2<br>Detailed info (or marked "normal" or similar) |
| | *One point if any info. Two points if statement summarizing body map (e.g., "exam consistent with sexual assault") or marked "normal" or similar.* | | |
| 11. Date of last consensual intercourse | | 0<br>Not present | 2<br>Present |
| | *Two points if date (or "n/a").* | | |
| **Forensic:** | | | |
| 12. Clothing info (4 fields) | 0<br>No fields completed | 1<br>Some fields completed | 2<br>Four fields completed |
| | *One point if 1–3 fields completed. Two points if ALL 4 fields completed (or "n/a").* | | |
| 13. Toilet and bathing info (2 fields) | 0<br>No fields completed | 1<br>One field completed | 2<br>Two fields completed |
| | *One point if 1 field completed. Two points if BOTH fields completed.* | | |
| 14. Info on perpetrator marks | | 0<br>No info | 2<br>Info reported |
| | *Two points if either box marked.* | | |
| **Genital examination:** | | | |
| 15. Genital exam info | 0<br>No info | 1<br>Some info | 2<br>Detailed info |

*(Continued)*

**Table 1.** (Continued)

| | Circle appropriate score | | |
|---|---|---|---|
| | *One point if any info. Two points if detailed info or marked "normal" or similar.* | | |
| 16. Statement in "Comments" summarizing genital exam | 0<br>No info | 1<br>Some info, or only stating hymen is intact/broken | 2<br>Detailed info |
| *One point if any info or if only discusses hymen. Two points if statement summarizing genital exam (e.g., "exam consistent with sexual assault") or marked "normal" or similar.* | | | |
| **Management**: | | | |
| 17. Management info | 0<br>Not present | | 2<br>Present |
| *Two points if any info or "n/a."* | | | |
| 18. Referral info | 0<br>Not present | | 2<br>Present |
| *Two points if any info or "n/a."* | | | |
| **Laboratory samples**: | | | |
| 19. Labs sent | 0<br>Not documented | | 2<br>Documented |
| *Two points if any documentation about labs (e.g., "none," "n/a," "HIV…," etc.).* | | | |
| **Chain of custody**: | | | |
| 20. List of chain-of-custody samples | 0<br>Not present | | 2<br>Present |
| *Two points if any documentation about chain-of-custody items (e.g., "none," "n/a," "clothing…," etc.).* | | | |
| 21. Examining Officer signature and date | 0<br>Not present | | 2<br>Present |
| *Two points if BOTH signature and date.* | | | |
| 22. Police Officer signature and date | 0<br>Not present | | 2<br>Present |
| *Two points if BOTH signature and date.* | | | |
| 23. Document signed by Examining Officer within 48 hours of patient visit | 0<br>Not signed within 48 hours | | 2<br>Signed within 48 hours |
| *Two points if signed within 48 hours of patient's visit.* | | | |
| **Psychological assessment (Part B)** | | | |
| 24. Part B (including child section if relevant) (NOTE: score is doubled for this checklist item) | 0<br>No info | 2<br>Some info | 4<br>Detailed info |
| *TWO points if any info. FOUR points if detailed info.* | | | |
| **General**: | | | |
| 25. Writing legible | 0<br>Not legible | 1<br>Partly legible | 2<br>Completely legible |
| *One point if partly legible. Two points if completely legible.* | | | |
| 26. Content understandable (e.g., clear meaning, avoids unexplained medical jargon, etc.) | 0<br>Not understandable | 1<br>Partly understandable | 2<br>Completely understandable |
| *One point if partly understandable. Two points if completely understandable.* | | | |
| **Total score**: Acceptable score = 43 (>80%) | | | **/54** |
| **Comments (specify checklist number followed by comment)**: | | | |

interventions to improve data quality and the overall response to sexual violence. Additionally, the Index could be used more broadly to accelerate Sustainable Development Goal 5.2, to end all forms of violence against women and girls [20].

There was perfect inter-rater agreement for many categorical and nominal variables on the sexual assault documentation forms, such as survivor demographics. There were lower

**Table 2. Inter-rater reliability results for each index item and for the PHR Data Quality Index overall.**

| | Scorer 1 | | Scorer 2 | | Mean score | Cohen's weighted Kappa (κ) | Interpretation of agreement* |
|---|---|---|---|---|---|---|---|
| **Demographics:** | **Mean** | **SD** | **Mean** | **SD** | | | |
| 1. All 4 dates (dates of form, birth, exam, incident) | 1.58 | 0.50 | 1.58 | 0.50 | 1.58 | 1.00 | Perfect |
| 2. Three names of survivor | 2.00 | 0.00 | 2.00 | 0.00 | 2.00 | 1.00 | Perfect |
| 3. Survivor contact info (address and phone) | 2.00 | 0.00 | 2.00 | 0.00 | 2.00 | 1.00 | Perfect |
| 4. OVC status | 1.48 | 0.89 | 1.48 | 0.89 | 1.48 | 1.00 | Perfect |
| **History:** | | | | | | | |
| 5. Perpetrator info (gender, est. age or adult/non-adult, unknown/known) | 1.87 | 0.50 | 1.87 | 0.50 | 1.87 | 1.00 | Perfect |
| 6. Chief complaints | 1.61 | 0.62 | 1.35 | 0.61 | 1.48 | 0.60 | Moderate |
| 7. Circumstances surrounding incident | 1.45 | 0.68 | 1.19 | 0.60 | 1.32 | 0.63 | Substantial |
| 8. Previous reporting and care | 1.81 | 0.60 | 1.81 | 0.60 | 1.81 | 1.00 | Perfect |
| **Physical examination:** | | | | | | | |
| 9. Notations on body map | 2.00 | 0.00 | 2.00 | 0.00 | 2.00 | 1.00 | Perfect |
| 10. Statement in 'Comments' summarizing body map exam | 1.61 | 0.72 | 1.74 | 0.68 | 1.68 | 0.76 | Substantial |
| 11. Date of last consensual intercourse | 0.84 | 1.00 | 0.84 | 1.00 | 0.84 | 1.00 | Perfect |
| **Forensic:** | | | | | | | |
| 12. Clothing info (4 fields) | 1.81 | 0.60 | 1.81 | 0.60 | 1.81 | 1.00 | Perfect |
| 13. Toilet and bathing info (2 fields) | 1.77 | 0.62 | 1.81 | 0.60 | 1.79 | 0.91 | Almost perfect |
| 14. Info on perpetrator marks | 1.81 | 0.60 | 1.81 | 0.60 | 1.81 | 1.00 | Perfect |
| 15. Genital exam info | 1.81 | 0.60 | 1.74 | 0.63 | 1.78 | 0.84 | Almost perfect |
| 16. Statement in 'Comments' summarizing genital exam | 1.45 | 0.81 | 1.42 | 0.81 | 1.44 | 0.96 | Almost perfect |
| **Management:** | | | | | | | |
| 17. Management info | 1.81 | 0.60 | 1.81 | 0.60 | 1.81 | 1.00 | Perfect |
| 18. Referral info | 1.81 | 0.60 | 1.81 | 0.60 | 1.81 | 1.00 | Perfect |
| **Laboratory samples:** | | | | | | | |
| 19. Labs sent | 1.81 | 0.60 | 1.81 | 0.60 | 1.81 | 1.00 | Perfect |
| **Chain of custody:** | | | | | | | |
| 20. List of chain-of-custody samples | 1.74 | 0.68 | 1.68 | 0.75 | 1.71 | 0.87 | Almost perfect |
| 21. Examining Officer signature and date | 1.77 | 0.62 | 1.68 | 0.75 | 1.73 | 0.80 | Substantial |
| 22. Police Officer signature and date | 0.13 | 0.50 | 0.13 | 0.50 | 0.13 | 1.00 | Perfect |
| 23. Document signed by Examining Officer within 48 hours of patient visit | 0.13 | 0.50 | 0.13 | 0.50 | 0.13 | 1.00 | Perfect |
| **Psychological assessment (Part B)** | | | | | | | |
| 24. Part B (including child section if relevant) (NOTE: score is doubled for this checklist item) | 3.61 | 1.20 | 3.61 | 1.20 | 3.61 | 1.00 | Perfect |
| **General:** | | | | | | | |
| 25. Writing legible | 2.00 | 0.00 | 2.00 | 0.00 | 2.00 | 1.00 | Perfect |
| 26. Content understandable (e.g., clear meaning, avoids unexplained medical jargon, etc.) | 1.61 | 0.62 | 1.71 | 0.59 | 1.66 | 0.82 | Almost perfect |
| **Total score out of 54:** | **43.32** | **11.02** | **42.81** | **10.86** | **43.07** | **0.77** | **Substantial** |
| Acceptable score = 43 (>80%) | | | | | | | |

*Interpretation of Cohen's Weighted Kappa: Poor agreement, 0.00; slight agreement, 0.00–0.20; fair agreement, 0.21–0.40; moderate agreement, 0.41–0.60; substantial agreement, 0.61–0.80; almost perfect agreement, 0.81–0.99; perfect agreement, 1.00 [19].

agreement rates for more subjective items, such as chief complaint and circumstances surrounding the sexual assault incident. However, not all subjective variables had poor agreement; in fact, scoring of the psychological assessment and legibility both had perfect agreement. This suggests that more subjective variables with lower inter-rater agreement may have the capacity

**Table 3. Summary of inter-rater reliability by level of agreement.**

| Interpretation of agreement (Kappa score range) | No. of items (%) |
|---|---|
| | (N = 26) |
| Perfect (1.00) | 17 (65.4%) |
| Almost perfect (0.81–0.99) | 5 (19.2%) |
| Substantial (0.61–0.80) | 3 (11.6%) |
| Moderate (0.41–0.60) | 1 (3.8%) |

to improve rating scoring either through adjustment of the variables or improved user guidance. As this quality assessment tool is implemented in the Kenyan context, researchers will continue to evaluate how lower-scoring measures can be optimized for improved agreement. The PHR team plans to utilize the validated Index to compare quality assessments between post-sexual assault paper-based forms to those collected via the MediCapt app, a digital form platform.

This present Index was developed for Kenyan medical professionals; however, it highlights the need to develop similar validated data quality indices for sexual assault documentation in other parts of the world. Stakeholders in the assessment of sexual assault in other contexts may review the validated tool to assess its applicability to other widely used forms and their unique environment and consider adaptation for implementation in their health care facilities. Drawing from sexual assault research such as the present study, sexual assault experts should identify global, standardized measures for high-quality sexual assault documentation and develop a validated global standard for quality assessment of sexual assault documentation that could be adapted to local needs and forms.

The present PHR Data Quality Index could be adapted for use in multiple contexts, such as future sexual violence research, health professional training, program evaluation, and targeted quality improvement post-training interventions. Research may be performed to test its validation in other contexts and to identify which documentation measures could be enhanced or added. The Index could be used for health professional education, including undergraduate, graduate, and continuing education to improve sexual assault examination and documentation. Additionally, sexual assault programs may choose to use the Index to assess the quality of the current sexual assault documentation practices, and target weaknesses and thereby enhance quality of documentation and, consequently, care for survivors.

The study has multiple strengths. The PHR Data Quality Index included feedback from several global and local sexual violence clinicians and human rights experts as well as, most importantly, experienced Kenyan health care professionals who use the sexual assault forms in

**Table 4. Summary of mean rater data quality scores for item data reported on the PRC Form.**

| Mean data quality score (out of a maximum of 2) | No. of items (%) |
|---|---|
| | (N = 26) |
| >1.5–2 | 19 (73.1%)* |
| >1.0–1.5 | 4 (15.4%) |
| >0.5–1.0 | 1 (3.8%) |
| 0–0.5 | 2 (7.7%) |

*Item #24, which is related to Part B of the PRC Form, had a maximum potential score of 4, received a mean rater score of 3.61 (or 1.81 if adjusted to a maximum score of 2), and is included in the cell indicated.

the field. The Index went through several iterations of review by multidisciplinary professionals before the final Index was determined. Inter-rater reliability testing of the Index showed substantial agreement overall.

There are limitations to this study, including the use of the kappa coefficient. While it is commonly used in statistics, some researchers argue it may be too lenient for health-related studies [21]. To address this intrinsic limitation of the kappa statistic, we included percent agreement alongside kappa coefficients, as suggested by several health services researchers [21]. An additional limitation of the study is its external validity, as it was developed using Kenyan post-sexual assault forms and may not be generalizable to other contexts and geographies. Future validation studies should include indices specific to the sexual assault forms from other geographies. Lastly, it is important to consider that the PHR Data Quality Index does not, in its current form, make a broad-scope assessment of external data accuracy.

## Conclusion

This study reports the development of a novel data quality index for sexual assault documentation. The index had substantial reliability, making it the first published validated quality-assessment tool for sexual assault documentation. The high inter-rater reliability suggests that the Index may be a promising strategy to enhance the quality of sexual assault documentation in other countries, with the goal of improving health care and justice for survivors.

## Supporting information

**S1 Fig.**
(PDF)

## Author Contributions

**Conceptualization:** Roseline Muchai, Katy Johnson, Ranit Mishori, Brett Nelson.

**Data curation:** Wendy Macias-Konstantopoulos, Brett Nelson.

**Formal analysis:** Rose McKeon Olson, Brett Nelson.

**Investigation:** Brett Nelson.

**Methodology:** Rose McKeon Olson, Wendy Macias-Konstantopoulos, Roseline Muchai, Brett Nelson.

**Project administration:** Roseline Muchai, Katy Johnson.

**Resources:** Wendy Macias-Konstantopoulos, Katy Johnson.

**Software:** Brett Nelson.

**Supervision:** Roseline Muchai, Ranit Mishori, Brett Nelson.

**Validation:** Rose McKeon Olson, Wendy Macias-Konstantopoulos, Brett Nelson.

**Visualization:** Brett Nelson.

**Writing – original draft:** Rose McKeon Olson.

**Writing – review & editing:** Rose McKeon Olson, Wendy Macias-Konstantopoulos, Roseline Muchai, Katy Johnson, Ranit Mishori, Brett Nelson.

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
