## [Decision Letter · Decision Letter 0]

9 Nov 2021

PONE-D-21-17812Development and validation of a data quality index for forensic documentation of sexual and gender-based violence in KenyaPLOS ONE

Dear Dr. Olson,

Thank you for submitting your manuscript to PLOS ONE. After careful consideration, we feel that it has merit but does not fully meet PLOS ONE’s publication criteria as it currently stands. Therefore, we invite you to submit a revised version of the manuscript that addresses the points raised during the review process.

We look forward to receiving your revised manuscript.

Kind regards,

Nancy Beam, PhD

Staff Editor

PLOS ONE

On behalf of:

Michelle L. Munro-Kramer, PhD, CNM, FNP-BC

Academic Editor

Journal Requirements:

Additional Editor Comments (if provided):

Thank you for this important manuscript. I apologize about the delay in review time. I tried repeatedly to get more than one reviewer with no success. Both myself and Reviewer #1 agree that this is a very well-written and important manuscript. Reviewer #1 has recommended some minor edits to the reference list (which I agree with) and should be undertaken before publication. Thank you for your important work.

Reviewers' comments:

Reviewer's Responses to Questions

**Comments to the Author**

1. Is the manuscript technically sound, and do the data support the conclusions?

Reviewer #1: Yes

2. Has the statistical analysis been performed appropriately and rigorously? 

Reviewer #1: Yes

3. Have the authors made all data underlying the findings in their manuscript fully available?

Reviewer #1: Yes

4. Is the manuscript presented in an intelligible fashion and written in standard English?

Reviewer #1: Yes

5. Review Comments to the Author

Reviewer #1: All seem to be in order. There are few errors in Reference list. Reference No.5: "Full Article" is indicated as name of author. Please find real authors for accurate referencing. Nu6: It is unusual to start with the year. Number. 12 does not seem to be in sync with other references, revise referencing style.

6. PLOS authors have the option to publish the peer review history of their article (what does this mean?). If published, this will include your full peer review and any attached files.

Reviewer #1: **Yes: **Prof Sinegugu E. Duma

---

## [Author Response · Author response to Decision Letter 0]

14 Nov 2021

Response to Reviewer #1: 

Thank you very much for your review and we agree with your suggestion. We have reviewed all references and ensured they are now accurate per the Vancouver reference style.

Response to Editor Comments:

Thank you for your comments and we appreciate your efforts and enthusiasm. We have ensured all references are now in accordance with the Author Guidelines and accurately reported.

---

## [Editor Report · Decision Letter 1]

21 Dec 2021

Development and validation of a data quality index for forensic documentation of sexual and gender-based violence in Kenya

PONE-D-21-17812R1

Dear Dr. Olson,

We’re pleased to inform you that your manuscript has been judged scientifically suitable for publication and will be formally accepted for publication once it meets all outstanding technical requirements.

Kind regards,

Michelle L. Munro-Kramer, PhD, CNM, FNP-BC

Academic Editor

PLOS ONE

Additional Editor Comments (optional):

Thank you for making all suggestions changes. Congratulations on your accepted manuscript.
---

## [Editor Report · Acceptance letter]

13 Jan 2022

PONE-D-21-17812R1 

Development and validation of a data quality index for forensic documentation of sexual and gender-based violence in Kenya 

Dear Dr. Olson:

I'm pleased to inform you that your manuscript has been deemed suitable for publication in PLOS ONE. Congratulations! Your manuscript is now with our production department. 

Kind regards, 

on behalf of

Dr. Michelle L. Munro-Kramer 

Academic Editor

PLOS ONE